biomimetics/robotics/biomechanics

locomotion, complex terrain, terradynamics, contact, deformation, robophysics

**Author for correspondence:**
Chen Li
e-mail: chen.li@jhu.edu

# Robotic modelling of snake traversing large, smooth obstacles reveals stability benefits of body compliance

## Qiyuan Fu and Chen Li

Department of Mechanical Engineering, Johns Hopkins University, Baltimore, MD, USA

 QF, 0000-0002-5275-4555;  CL, 0000-0001-7516-3646

Snakes can move through almost any terrain. Although their locomotion on flat surfaces using planar gaits is inherently stable, when snakes deform their body out of plane to traverse complex terrain, maintaining stability becomes a challenge. On trees and desert dunes, snakes grip branches or brace against depressed sand for stability. However, how they stably surmount obstacles like boulders too large and smooth to gain such 'anchor points' is less understood. Similarly, snake robots are challenged to stably traverse large, smooth obstacles for search and rescue and building inspection. Our recent study discovered that snakes combine body lateral undulation and cantilevering to stably traverse large steps. Here, we developed a snake robot with this gait and snake-like anisotropic friction and used it as a physical model to understand stability principles. The robot traversed steps as high as a third of its body length rapidly and stably. However, on higher steps, it was more likely to fail due to more frequent rolling and flipping over, which was absent in the snake with a compliant body. Adding body compliance reduced the robot's roll instability by statistically improving surface contact, without reducing speed. Besides advancing understanding of snake locomotion, our robot achieved high traversal speed surpassing most previous snake robots and approaching snakes, while maintaining high traversal probability.

## 1. Introduction

Snakes are masters of locomotion across different environments [1]. With their elongate, flexible body [2] of many degrees of freedom [3], snakes can use various planar gaits to move on flat surfaces, be it open [4–6], confined [4–6] or with small obstacles that can be circumvented [5,7]. Snakes can also deform their body out of plane to move across complex environments [8–12]

**Figure 1.** A snake combines body lateral undulation and cantilevering to traverse a large step stably [12]. Representative snapshots of a kingsnake traversing a large step (oblique view) with the base of support and centre of mass overlaid. (*a*) Before cantilevering. (*b*) During cantilevering but before reaching upper surface. (*c*) After reaching upper surface. (*d*) Lifting off lower surface. (*e*) After the entire body reaches upper surface. In each snapshot, yellow curve shows body midline, red polygon shows base of support formed by body sections in contact with horizontal surfaces, white point shows centre of mass and white circle and dashed line show the projection of the centre of mass onto a horizontal surface.

(for a review, see [12] electronic supplementary material). In these situations, out-of-plane body deformation can challenge stable locomotion [11,13,14], which is rarely an issue on flat surfaces with planar gaits. To maintain stability, arboreal snakes grip or brace against branches [8,10,15,16] and carefully distribute body weight [16,17]; sidewinders depress portions of the body into sand and brace against it without causing avalanche, while minimizing out-of-plane deformation [11]. However, we still know relatively little about how snakes maintain stability when surmounting obstacles such as boulders that are too large and smooth to gain such 'anchor points' by gripping or bracing.

With a snake-like slender, reconfigurable body, snake robots hold the promise as versatile platforms to traverse diverse environments [18–20] for critical applications like search and rescue and building inspection [21,22]. Similar to snakes, snake robots are inherently stable when they use planar gaits on flat surfaces [23,24] but face stability challenges when they deform out of plane in more complex environments [11,13,14,25–27]. In branch-like terrain and confined spaces and on sandy slopes, snake robots also maintain stability by gripping or bracing against the surfaces or depressed sand [11,28–30]. Surmounting large, smooth obstacles like steps has often been achieved using a simple, follow-the-leader gait [31–41], in which the body deforms nearly within a vertical plane with little lateral deformation and hence a narrow base of ground support. Only two previous snake robots deliberately used lateral body deformation for a wide base of support when traversing large steps [28,39,40]. Regardless, all these previous snake robots rely on the careful planning and control of motion to maintain continuous static stability and thus often traverse at low speeds. Better understanding of the stability challenges of high-speed locomotion over large, smooth obstacles can help snake robots traverse more rapidly and stably.

In a recent study [12], our group studied the generalist kingsnake traversing large steps by partitioning its body into sections with distinct functions (figure 1). The body sections below and above the step always undulate laterally on horizontal surfaces to propel the animal forward, while the body section in between cantilevers in the air in a vertical plane to bridge the height increase. An important insight was that lateral body undulation helps maintain stability by creating a wide base of ground support to resist lateral perturbations (figure 1, red regions). Without it, when a long body section is cantilevering in the air but has not reached the upper surface (figure 1*b*), a significant roll perturbation can tip the animal over. Thanks to body partitioning with lateral undulation, the snake traversed steps as high as 25% body length (or 30% snout-vent length) with perfect stability [12]. A signature of its perfect stability was that the laterally undulating body sections never lifted off involuntarily from horizontal surfaces before and after cantilevering.

In this study, we take the next step in understanding the stability principles of large step traversal using lateral undulation combined with cantilevering, by testing two hypotheses: (1) roll stability diminishes as step becomes higher and (2) body compliance improves surface contact statistically and reduces roll instability. The kingsnake did not attempt to traverse steps higher than 25% body length (on which it maintains perfect stability) and their body compliance cannot be modified without affecting locomotion. Thus, to test these two hypotheses, we developed a snake robot as a physical model which we could challenge with higher steps and whose body compliance could be modified. The second hypothesis was motivated by the observation during preliminary experiments that the

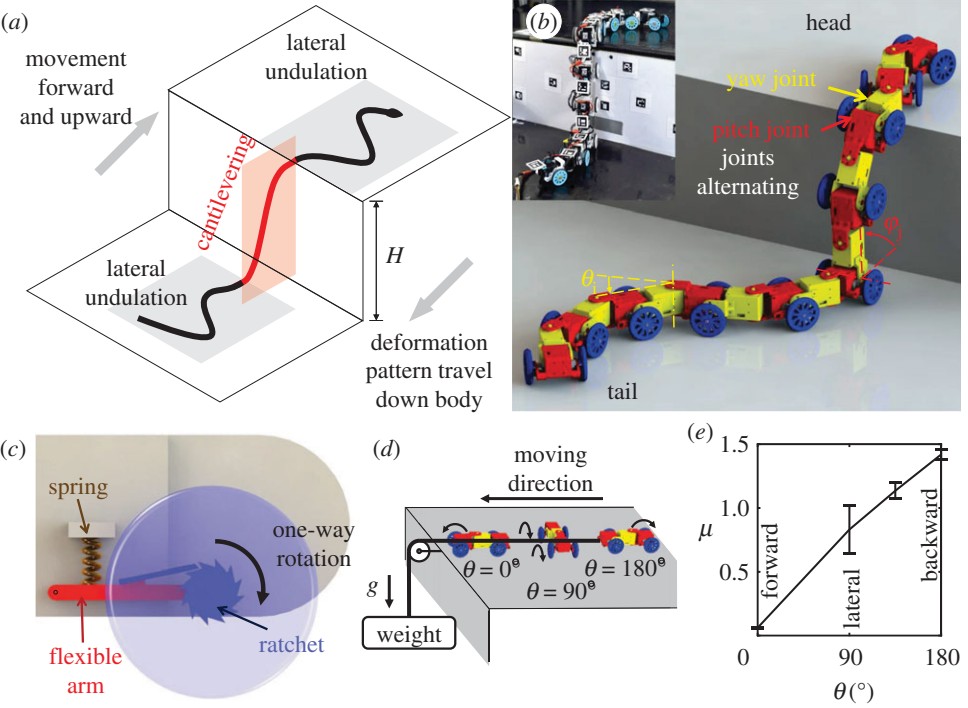

**Figure 2.** Gait and mechanical design of snake robot. (*a*) Partitioned gait template from kingsnakes combining lateral undulation and cantilevering to traverse a large step [12]. Lateral undulation can be simply controlled and varied using a few wave parameters, such as wavelength, amplitude and frequency. (*b*) The snake robot consists of serially connected segments with alternating pitch (red) and yaw (yellow) joints and one-way wheels (blue). (*c*) Close-up view of a one-way wheel (blue) attached to each pitch segment, which only rotates forward via a ratchet mechanism. To add mechanical compliance, wheel connects to body segment via a suspension system with a spring (brown) and a flexible arm (red). Suspension is disabled in rigid robot experiments by inserting a lightweight (0.4 g) block with the same length as natural length of spring. (*d*) Experimental set-up to measure kinetic friction coefficient. Three segments are dragged by a weight using a string through a pulley with various orientation angle $\theta$ between body long axis and direction of drag force. $\theta = 0°$, 90° and 180° are for sliding in body's forward, lateral and backward directions, respectively. Arrow on one-way wheel shows its direction of free rotation. (*e*) Kinetic friction coefficient as a function of body orientation $\theta$. Error bars show ± 1 s.d. See electronic supplementary material, movie S1 for demonstration of robot mechanisms.

robot with a rigid body often rolled to the extent of involuntary lift-off from horizontal surfaces, in contrast with the snake with compliant body [2] that never did so (see §2.5 for details).

# 2. Physical modelling with rigid snake robot

## 2.1. Mechanical design

Our snake robot used the partitioned gait template (figure 2*a*) from our recent animal observations [12]. The robot was 107 cm long, 8.2 cm tall and 6.5 cm wide and weighed 2.36 kg excluding off-board controllers and extension cables. To enable large body deformation both laterally and dorsoventrally for traversing large steps (and complex three-dimensional terrain in general), the robot consisted of 19 segments with 19 servo-motors connected by alternating pitch (red) and yaw (yellow) joints (figure 2*b*; electronic supplementary material, movie S1, see details in electronic supplementary material), similar to [42]. We refer to segments containing pitch or yaw joint servo-motors as pitch or yaw segments, respectively.

An anisotropic friction profile, with smaller forward friction than backward and lateral friction, is critical to snakes' ability to move on flat surfaces using lateral undulation [43]. To achieve this in the robot, we added to each pitch segment a pair of one-way wheels (48 mm diameter, with a rubber O-ring on each wheel) realized by a ratchet mechanism similar to [44] (figure 2*b,c*, blue; electronic supplementary material, movie S1). The one-way wheels unlocked when rotating forward and locked when rotating backward, resulting in a small forward rolling friction and a large backward sliding

friction, besides a large lateral sliding friction. We measured the kinetic friction coefficient at various body orientation (figure 2$d$; see details in electronic supplementary material) and confirmed that forward friction was indeed smaller than backward and lateral friction (figure 2$e$).

## 2.2. Control of body lateral undulation and cantilevering

To generate lateral undulation on the robot's body sections below and above the step, we applied a serpenoid travelling wave spatially on the body shape with sinusoidal curvature [18] in the horizontal plane, which propagated from the head to the tail. The wave form, with a wavenumber of 1.125, was discretized onto the robot's yaw segments in these two sections, by actuating each yaw joint to follow a temporal sinusoidal wave with an amplitude of 30°, a frequency of 0.25 Hz and a phase difference of 45° between adjacent yaw joints. The travelling wave form in the section below the step immediately followed that above, as if they formed a single wave, if the cantilevering section was not considered. We chose a serpenoid travelling wave because it is similar to that used by kingsnakes and easy to implement in snake robots [18,45,46]. We chose wave parameters from preliminary experiments and kept them constant in this study to study the effect of step height and body compliance.

To generate cantilevering on the section in between, for each step height tested, we used the minimal number of pitch segments required to bridge across the step. The cantilevering section was kept straight and as vertical as possible, except that the two most anterior pitch segments pitched forward for the anterior undulating body section to gain proper contact with the upper surface (electronic supplementary material, figure S2A, see details in electronic supplementary material). This shape was calculated based on the step height measured from online camera tracking before body cantilevering started and remained the same while travelling down the body. Overall, with this partitioned gait template, control of the robot's many degrees of freedom was simplified to using only a few wave parameters.

To propagate the three partitioned sections down the robot as it moved forward and upward onto the step, we used the measured positions of the robot's segments to conduct feedback logic control (electronic supplementary material, figure S2C) similar to [40]. An online camera tracked ArUco markers attached to each pitch segment and the step surfaces, and the distance of each segment relative to the step in the forward and upward direction was calculated. This distance was used to determine when each segment should transition from lateral undulation to cantilevering or conversely. Below we refer to this process as section division propagation. See more technical details of robot control in electronic supplementary material.

Apart from the experimenter starting and stopping it, the robot's motion to traverse the step was automatically controlled by a computer (electronic supplementary material, figure S2B). The experimenter stopped the robot when it: (i) flipped over, (ii) became stuck for over 10 undulation cycles or (iii) traversed the step.

## 2.3. Traversal probability diminishes as step becomes higher

To test our first hypothesis, we challenged the robot to traverse increasingly large, high friction step obstacles (electronic supplementary material, figure S1A), with step height $H = 33, 38, 41$ and $43$ cm, or 31, 36, 38 and 40% of robot length $L$ (see representative trial in figure 3$a$; electronic supplementary material, movie S2 and S3, left). Using body lateral undulation combined with cantilevering, the robot traversed a step as high as near a third of body length (31% $L$) with a high probability of 90% (figure 3$c$, black dashed). In addition, its motion during traversal was more dynamic than previous snake robots that traverse steps using quasi-static motion (electronic supplementary material, movie S2 and S3, left). However, as it attempted to traverse higher steps, the robot struggled (electronic supplementary material, movie S3, left) and its traversal probability quickly decreased ($p < 0.005$, simple logistic regression), diminishing to 20% when step height reached 40% $L$.

## 2.4. Poorer roll stability on higher steps increases failure

To determine whether the diminishing traversal probability was caused by diminishing roll stability, we recorded high-speed videos of all experimental trials (electronic supplementary material, figure S1B). Observation of these videos revealed that failure to traverse was a result of one or a sequence of adverse events (figures 4 and 5, electronic supplementary material, movie S4).

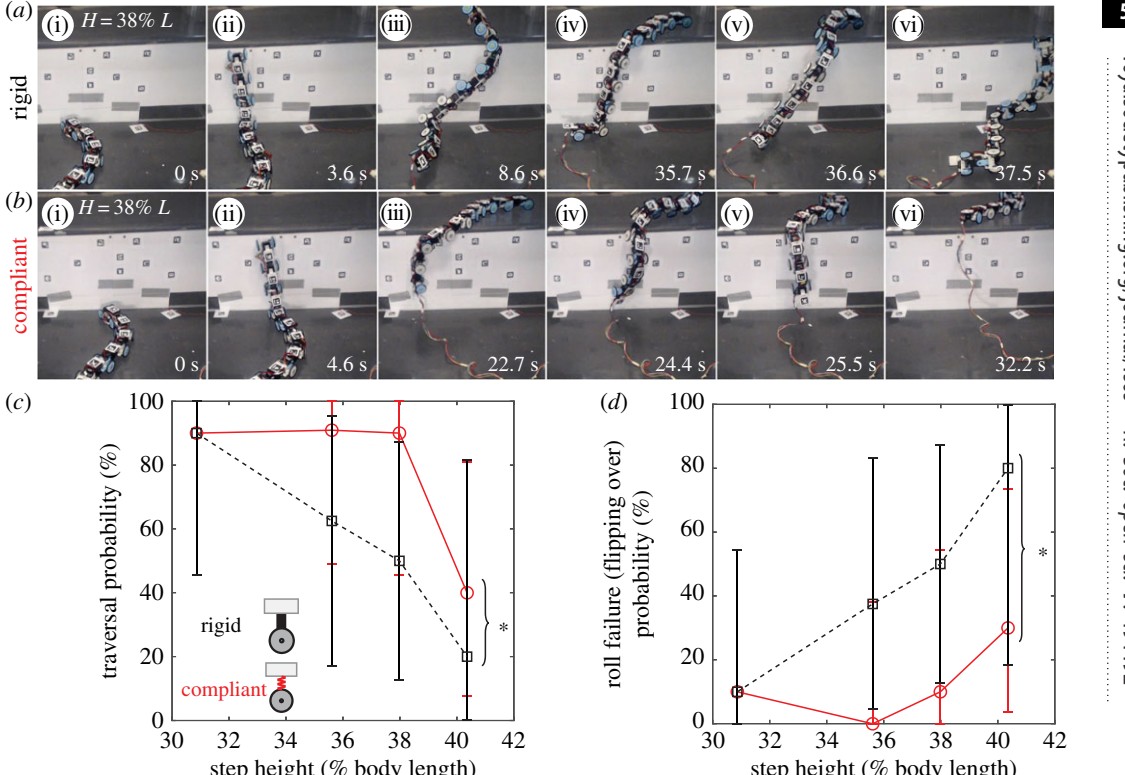

**Figure 3.** Traversal performance of robot. Representative snapshots of robot with a rigid body (*a*) and a compliant body (*b*) traversing a step as high as 38% body length. Body rolling back and forth (wobble) is visible in (*a*) iii–vi and (*b*) iii–v. Rigid body robot failed to recover from rolling and eventually flipped over; see (*a*) vi and electronic supplementary material, movie S3, left, for a representative video. Rolling is less severe for compliant body robot, which often recovers or transitions to other events and rarely flips over; see (*b*) vi and electronic supplementary material, movie S3, right, for a representative video. (*c*) Traversal probability as a function of step height. Bracket and asterisk show a significant difference between rigid and compliant body robot ($p < 0.05$, multiple logistic regression). (*d*) Effect of body compliance on probability of roll failure (i.e. flipping over, see §2.4). Bracket and asterisk represent a significant difference between rigid and compliant body robot ($p < 0.005$, multiple logistic regression). In (*c*) and (*d*), black dashed is for rigid body robot; red solid is for compliant body robot. Error bars show 95% confidence intervals.

(1) Imperfect lift timing (figure 4*a*,*b*; figure 5, grey). This includes lifting too early (figure 4*a*) or late (figure 4*b*) due to inaccurate estimation of body forward position relative to the step. Noise in the system, both mechanical (e.g. variation in robot segment and surface friction) and in feedback control (e.g. camera noise, controller delay), resulted in trial-to-trial variation of robot motion and interaction with the step, leading to inaccurate position estimation. With underestimation, a segment still far away from the step was lifted too early (figure 4*a*, segment between i and i + 1). With overestimation, a segment close to the step was lifted too late and pushed against the step (figure 4*b*, segment between i and i + 1). These control imperfections often triggered other adverse events stochastically, as described below.

(2) Stuck (figure 4*c*; figure 5, yellow). The robot was occasionally stuck when its cantilevering section pushed against the step with no substantial body yawing or rolling. After becoming stuck, the robot always eventually recovered within 10 undulation periods and succeeded in traversing (figure 5, no purple arrows).

(3) Yawing (figure 4*d*; figure 5, blue). The robot often yawed substantially when the sections below and/ or above step slipped laterally. This was always triggered by imperfect lift timing (figure 5, arrows from grey to blue box) which led the cantilevering section to push against the vertical surface (even with lifting too early, the extra weight suspended in the air sometimes resulted in sagging of the cantilevering section, which in turn pushed against the vertical surface). The push resulted in a yawing torque too large to be overcome by the frictional force on the undulating sections from the horizontal surfaces. Because of this, yawing was often accompanied by small lift-off and slip of the undulating segments, which could lead to rolling (figure 5, blue arrows), described below.

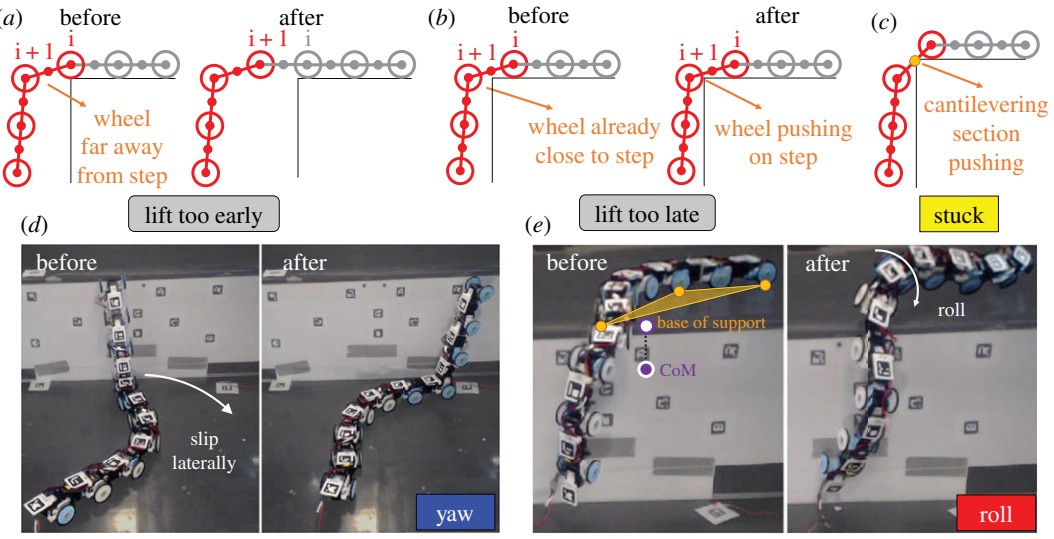

**Figure 4.** Adverse events leading to failure. (*a*) Lift early: robot lifts up a segment too early before preceding segment moves onto the upper surface. (*b*) Lift late: robot lifts up a segment too late and a cantilevering segment closest to top edge of step pushes against it. In (*a* and *b*), indices above wheels are from head to tail. (*c*) Stuck: robot becomes stuck when cantilevering section pushes against step, with no overall direction or position change. This happens when robot belly instead of wheels contacts top edge of step. In (*a*–*c*), red and grey sections are cantilevering and undulating sections. (*d*) Yaw: robot yaws due to lateral slipping of body sections below and/or above step. (*e*) Roll: robot rolls with the loss of contact below and/or above step; purple and white points are the centre of mass (CoM) and its projection onto the upper horizontal surface. In (*c*) and (*e*), orange points are contact points, and orange shade is base of support. See electronic supplementary material, movie S4 for examples of (*c*–*e*).

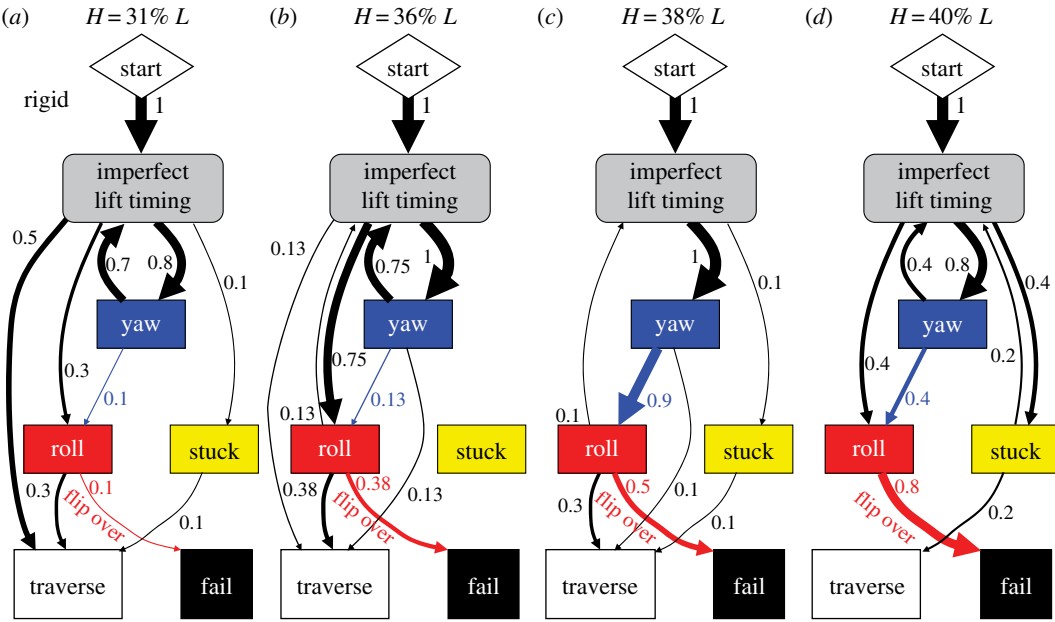

**Figure 5.** Transition pathways of rigid body robot among adverse events to traverse or fail. (*a*) Step height *H* = 31% robot length *L*. (*b*) *H* = 36% *L*. (*c*) *H* = 38% *L*. (*d*) *H* = 40% *L*. Each arrow is a transition between nodes, with arrow thickness proportional to its probability of occurrence, shown by number next to it. Probability of occurrence here is the ratio of the number of trials in which a transition occurs to the total number of trials; it is different from transition probability in Markov chains. If a transition occurs multiple times in a trial, it is only counted once.

Yawing could also compromise segment position estimation and sometimes led to further imperfect lift timing (figure 5, arrows from blue to grey box).

(4) Rolling (figure 4*e*; figure 5, red). The robot rolled about the fore–aft axis substantially as the centre of mass (figure 4*e*, purple point) projection (white point) moved out of the base of support (orange

shade). Sometimes, this instability was induced by a sudden shift of centre of mass position during segment lifting. At other times, this instability was induced by sudden shrinking of base of support due to loss of surface contact, which resulted from either small lift-off and/or slip of segments due to yawing or the last segment lifting off the lower surface. When rolling occurred, the robot suddenly lost many contacts from the horizontal surfaces and often lost thrust and stability. Sometimes the robot could roll back by its own bending motion (figure 3b iv and v). If not, it would flip over (and sometimes fell off the step) (figure 3a, v and vi; electronic supplementary material, movie S4), resulting in failure to traverse (figure 5, red arrows). Hereafter, we refer to the robot flipping over due to rolling as roll failure.

For all step heights tested, we observed a diversity of pathways stochastically transitioning among these adverse events (figure 5). Given the diverse, stochastic transitions, statistical trends emerged in their pathways. First, failure was always directly resulting from rolling to the extent of flipping over, i.e. roll failure (figure 5, red arrows). In addition, as step height increased from 31% $L$ to 40% $L$, roll failure (flipping over) became more likely, from 10 to 80% probability (figure 5a–d, red arrows; figure 3d, black dashed; $p < 0.05$, simple logistic regression), which resulted in decreasing traversal probability from 90 to 20% (figure 3c, black dashed). This confirmed our first hypothesis that increasing step height diminishes roll stability. This diminishing was a direct result of the shorter undulating body sections for lateral support as the cantilevering body section lengthened as step height increased.

## 2.5. Comparison with snakes

Comparison of robot with animal observations [12] revealed and elucidated the snake's better ability to maintain stability over the robot. First, the robot always suffered imperfect body lift timing (figure 5, arrows from start to grey box), which was rarely observed in the snake [12]. Second, the robot's laterally undulating body sections often suffered large yawing ($\geq$ 80% probability, figure 5, blue) and rolling ($\geq$ 40%, figure 5, red). By contrast, the snake's undulating body sections rarely rolled on the horizontal surfaces, even when step friction was low and the animal slipped substantially [12]. Third, when step friction was high, the robot sometimes became stuck (figure 5, yellow), whereas the snake always smoothly traversed [12]. These indicate that the snake is better at accommodating noise and perturbations in control, design and the environment (e.g. improper timing, unexpected forces and slippage, variation in step surface height and friction) to maintain effective body–terrain interaction.

Besides the animal's better ability to use sensory feedback to control movement in complex environments [47], the two morphological features of the snake body probably contributed to its better ability to maintain stability—being more continuous (over 200 vertebrae [3] versus the robot's 19 segments) and more compliant [2]. The latter is particularly plausible considering that the introduction of mechanical compliance to end effectors has proven crucial in robotic tasks where contact with the environment is essential, e.g. grasping [48–50], polishing [51] and climbing [52,53] robots (for a review, see [54]). Although many snake robots have used compliance in control to adapt overall body shape to obstacles [55–58], the use of mechanical compliance to better conform to surfaces locally was less considered [56,59], especially for improving stability. These considerations inspired us to test our second hypothesis that body compliance improves surface contact statistically and reduces roll instability.

# 3. Physical modelling with compliant snake robot

## 3.1. Suspension to add mechanical compliance

To test our second hypothesis, we added mechanical compliance to the robot by inserting between each one-way wheel and its body segment a suspension system inspired by [59] (figure 2c). The suspension of each wheel (even the left and right on the same segment) could passively compress independently to conform to surface variation (by up to 10 mm displacement of each wheel). From our second hypothesis that body compliance improves surface contact statistically and reduces roll instability, we predicted that this passive conformation would increase traversal probability and reduce roll failure (flipping over) probability, especially for larger steps. The suspension system was present but disengaged in the rigid robot experiments for direct comparison.

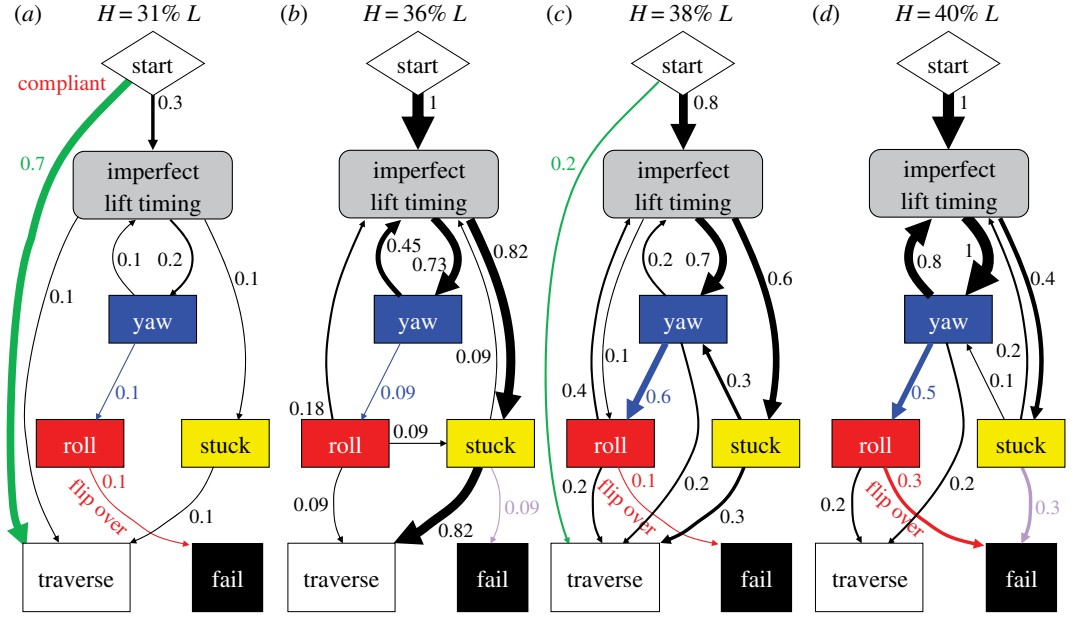

**Figure 6.** Transition pathways of compliant body robot among adverse events to traverse or fail. (*a*) Step height $H = 31\%$ robot length $L$. (*b*) $H = 36\%$ $L$. (*c*) $H = 38\%$ $L$. (*d*) $H = 40\%$ $L$. Figure 5 for definition of transition diagrams.

## 3.2. Body compliance increases traversal probability

The compliant body robot maintained high traversal probability over a larger range of step height than the rigid robot, consistently succeeding 90% of the time as step height increased from 31% $L$ to 38% $L$ (figure 3*c*, red solid). Like the rigid body robot, the compliant body robot's motion during traversal was also more dynamic than previous robots using quasi-static gaits (electronic supplementary material, movie S3, right). Traversal probability only decreased for step height beyond 38% $L$ ($p < 0.05$, pairwise Chi-square test). For the large step heights tested, adding body compliance increased traversal probability ($p < 0.05$, multiple logistic regression). These observations were in accord with our prediction from the second hypothesis.

## 3.3. Body compliance reduces roll failure probability

To test our second hypothesis, specifically that body compliance reduces roll instability, we compared the compliant body robot's transition pathways among adverse events (figure 6) to those of the rigid body robot (figure 5). The compliant body robot still stochastically transitioned among adverse events. However, two improvements were observed in the statistical trends of the transition pathways.

First, the compliant body robot suffered roll failure (flipping over) less frequently (figure 5 versus figure 6, red arrows; figure 3*d*; $p < 0.005$, multiple logistic regression), especially after yawing occurred (figure 5 versus figure 6, blue arrows). It also experienced less frequent back and forth rolling (wobbling) (figure 3*b*; electronic supplementary material, movie S3, right) than the rigid body robot (figure 3*a*; electronic supplementary material, movie S3, left). This confirmed our hypothesis that body compliance reduces roll instability. However, body compliance did not eliminate rolling, and the compliant robot still slipped frequently. These observations were in accord with our prediction from the second hypothesis.

Second, the compliant body robot suffered imperfect lift timing less frequently than the rigid body robot (figure 5 versus figure 6, green arrows), which eventually resulted in more frequent traversal for all step heights above 31% $L$ (figure 5 versus figure 6, sum of all black arrows into white box). This was because, even when lifting was too early or too late, the compliant body robot could often afford to push against the vertical surface without triggering catastrophic failure from yawing or rolling before section division propagation resumed.

However, the compliant body robot became stuck more frequently (figure 5 versus figure 6, arrow from grey to yellow box) and failed more frequently as a result (figure 5 versus figure 6, purple arrow). This was because compression of the suspension lowered ground clearance of the segments.

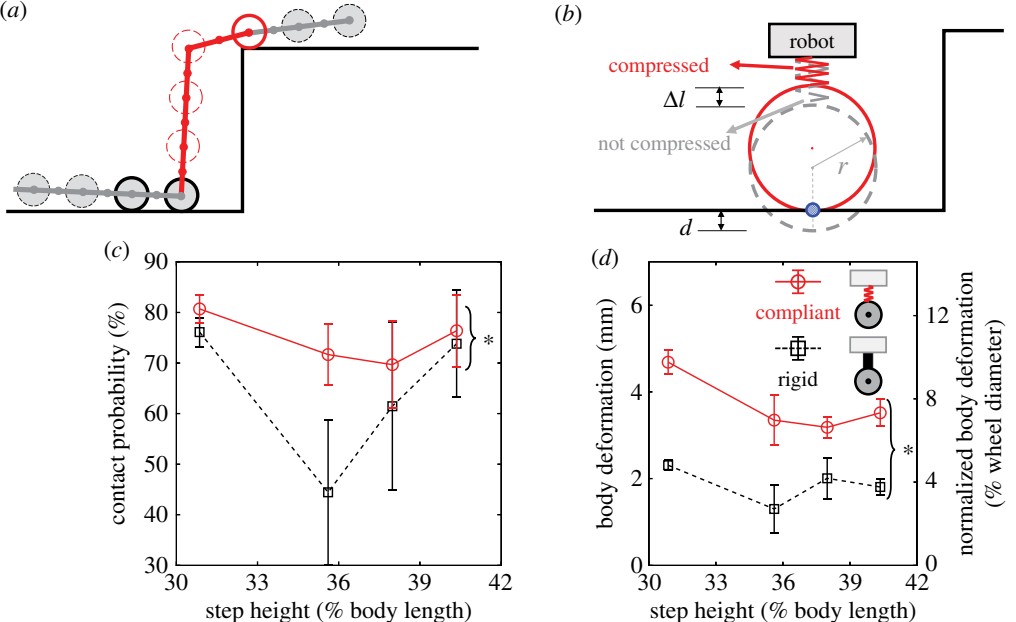

**Figure 7.** Effect of body compliance on contact probability and body deformation. (*a*) Example side view schematic to define contact probability. Grey are laterally undulating body sections and red is cantilevering body section. In this example, three wheels of undulating section are in contact with surface (solid) and four are not (dashed), and contact probability = 3/(3 + 4) = 43%. (*b*) Definition of body deformation. Red solid schematic shows actual wheel position with suspension compressed, and grey dashed one shows wheel position assuming no suspension compression. Blue circle shows point of contact between wheel and surface. Body deformation $\Delta l$ is approximated by virtual wheel penetration $d$ into surface. (*c*) Contact probability as a function of step height. (*d*) Body deformation as a function of step height. In (*c*) and (*d*), black dashed is for rigid body robot; red solid is for compliant body robot. Error bars show ± 1 s.d. Brackets and asterisks represent a significant difference between rigid and compliant body robot ($p < 0.001$, ANCOVA).

This stuck failure mode was always directly triggered by lifting too early. This is a limitation of the robot's discrete, few degree-of-freedom body.

## 3.4. Body compliance improves contact statistically

To further test our second hypothesis, specifically that body compliance improves surface contact statistically, we compared contact probability between the rigid and compliant body robot. Contact probability was defined as the ratio of the number of wheels contacting horizontal surfaces in the laterally undulating body sections to the total number of wheels in these two sections (figure 7*a*). In addition, we calculated body deformation as how much each wheel suspension was compressed for these two sections (figure 7*b*). Both wheel contact and body deformation were determined by examining whether any part of each wheel penetrated the step surface assuming no suspension compression, based on three-dimensional reconstruction of the robot from high-speed videos. Both were averaged spatio-temporally over the traversal process across all pitch segments in these two sections combined for each trial. See details in electronic supplementary material.

For all step heights tested, the compliant body robot had a higher contact probability than the rigid body robot (figure 7*c*; $p < 0.001$, ANCOVA). This improvement was a direct result of larger body deformation (figure 7*d*; $p < 0.0001$, ANCOVA): the rigid body robot only deformed around 2 mm or 4% wheel diameter (which occurred in the rubber on both the wheels and step surfaces); by contrast, the compliant robot's suspension deformed around 4 mm or 8% wheel diameter, a 100% increase.

## 3.5. Body compliance reduces severity of body rolling

Body rolling would result in lateral asymmetry in how the robot conforms with the surface between the left and right sides of the body. Thus, to quantify the severity of body rolling, we calculated the difference (absolute value) in surface conformation between left and right wheels (figure 8) excluding the cantilevering section. Surface conformation was defined as the virtual penetration of a wheel in

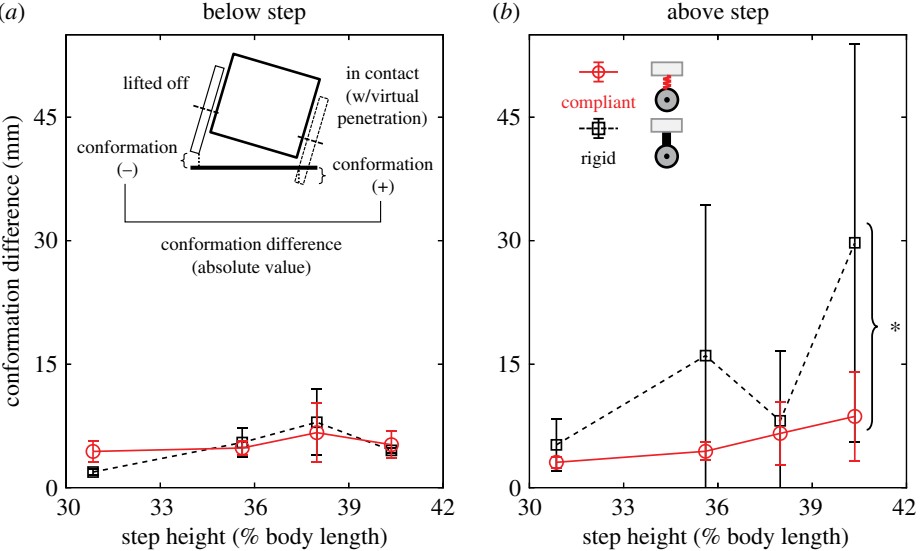

**Figure 8.** Effect of body compliance on surface conformation difference. Surface conformation difference between left and right wheels as a function of step height for body sections below (*a*) and above (*b*) step. Black dashed is for rigid body robot; red solid is for compliant body robot. Error bars show ± 1 s.d. Brackets and asterisks represent a significant difference between rigid and compliant body robot (*p* < 0.005, ANCOVA). Inset in (*a*) shows front view schematic to define surface conformation difference (see text for detail).

contact (positive distance) or the minimal distance from the surface of a wheel lifted off (negative distance) (figure 8*a* inset, right or left wheel). A larger difference (absolute value) in surface conformation between left and right sides means more severe rolling. Surface conformation difference was averaged spatio-temporally over the traversal process across all pitch segments separately for the body sections below and above the step for each trial. See details in electronic supplementary material.

For all step heights tested, body compliance reduced lateral asymmetry in surface conformation for the body section above the step (figure 8*b*; *p* < 0.005, ANCOVA), although not for the section below (figure 8*a*; *p* > 0.05, ANCOVA). This means that the compliant body robot had less severe body rolling above the step (figure 3*b*; electronic supplementary material, movie S3, right) and was more stable during traversal. Such better surface conformation probably allowed the compliant body robot to generate ground reaction forces more evenly along the body [60] to better propel itself forward and upward, which the rigid robot with poorer surface conformation struggled to do. The compliant body robot still wobbled and slipped more substantially than the snakes [12].

All these observations from the compliant body robot confirmed our second hypothesis that body compliance improves surface contact statistically and reduces roll instability.

## 3.6. Body compliance increases energetic cost

Not surprisingly, these benefits came with a price. The electrical power consumed by the robot increased when body compliance was added (electronic supplementary material, figure S3; *p* < 0.0001, ANCOVA; see details in electronic supplementary material). We speculate that this was due to an increase in energy dissipation from larger friction dissipation against the surfaces due to higher contact probability, viscoelastic response [54] of the suspension and more motor stalling and wheel sliding due to more frequently getting stuck. Electrical power consumption decreased with step height (electronic supplementary material, figure S3; *p* < 0.0001, ANCOVA), which may result from the decrease in the number of laterally undulating segments that dissipated energy during sliding against the surfaces. We noted that the electrical energy consumed (power integrated over time) during traversal was two orders of magnitude larger than the mechanical work needed to lift the robot onto the step; the majority of the energy was not used to do useful work [61].

# 4. Contribution to robotics

Our study advanced the performance of snake robots traversing large steps. When normalized to body length, our robot (both rigid and compliant body, figure 9, red and black) achieved step traversal speed

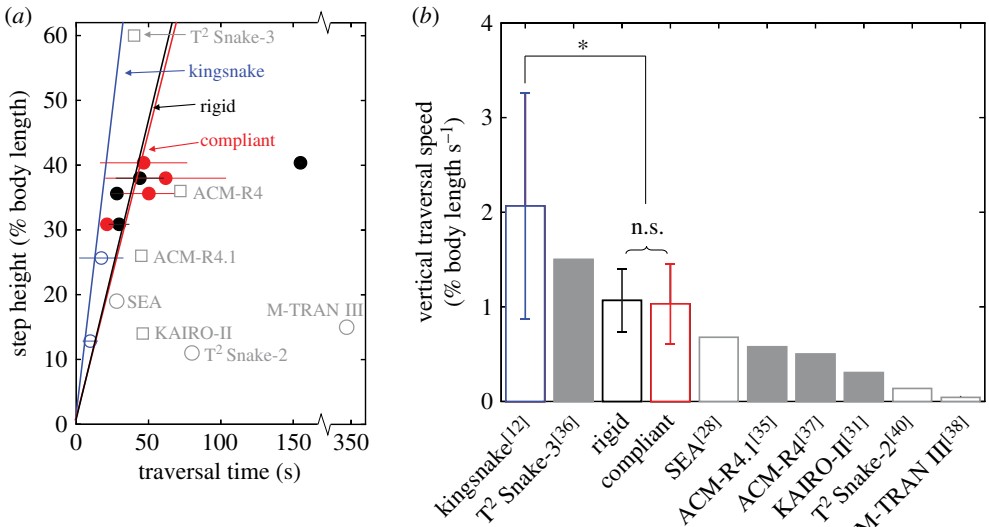

**Figure 9.** Comparison of traversal performance of our robot with previous snake robots and the kingsnake. (*a*) Maximal traversable step height (normalized to body length) as a function of traversal time for kingsnake (blue), our robot with rigid (black) and (red) compliant body, and previous snake robots with data available (grey squares: with active propellers; grey circles: no active propellers). Several previous robots with no traversal time reported are not included [21,33,34,42,62,63]. (*b*) Vertical traversal speed normalized to body length. Vertical traversal speed, i.e. normalized step height divided by traversal time, is the slope of lines connecting each data point to the origin in (*a*). Thus, a higher slope indicates a larger vertical traversal speed. Speeds of previous robots are the fastest reported values (vs. average in ours) from [28,31,35–38,40] or accompanying videos. See electronic supplementary material for details of speed calculation. Bracket and asterisk represent a significant difference in vertical traversal speed ($p < 0.005$, pairwise two-sample $t$-test). n.s. represents no significant difference ($p > 0.05$, pairwise two-sample $t$-test). In (*a,b*), error bars show ± 1 s.d.

higher than most previous snake robots (grey) and approaching that of kingsnakes (blue). In addition, our compliant robot maintained high traversal probability (90%) even when the step was as high as 38% body length (figure 3*c*, red solid), without loss of traversal speed compared to the rigid body robot ($p > 0.05$, pairwise two-sample $t$-test). These improvements were attributed to the inherent roll stability from body lateral undulation and an improved ability to maintain surface contact via body compliance, which alleviated the need for precise control for static stability and enabled dynamic traversal. The only other snake robot that achieved comparable step traversal speed (T$^2$ Snake-3 [36]) used active wheels to drive the body while only deforming in a vertical plane and thus had low roll stability. It also pushed against and propelled its wheels up the vertical surface, which reduced the load of pitching segments and increased the maximal cantilevering length. We note that our robot still has the potential to achieve even higher speeds with high traversal probability, because in our experiments, the motors were actuated at only 50% full speed to protect the robot from breaking after drastically flipping over as well as motor overload due to inertial forces and we have yet to systematically test and identify optimal serpenoid wave parameters [64].

## 5. Summary and future work

Inspired by our recent observations in snakes, we developed a snake robot as a physical model and performed systematic experiments to study stability principles of large step traversal using a partitioned gait that combines lateral body undulation and cantilevering. Our experiments confirmed two hypotheses: (1) roll stability diminishes as step becomes higher and (2) body compliance improves surface contact statistically and reduces roll instability. In addition, thanks to the integration of lateral body undulation to resist roll instability with anisotropic friction for thrust, our snake robot traversed large step obstacles more dynamically than previous robots with higher traversal speeds (normalized to body length), approaching animal performance. Moreover, by further adding body compliance to improve surface contact, our snake robot better maintained high traversal probability on high steps without loss in traversal speed. Although our discoveries were made on a simple large step with only vertical body compliance, the use of body lateral undulation and compliance to

achieve a large base of support with reliable contact for roll stability is broadly useful for snakes and snake robots traversing other large, smooth obstacles in terrain like non-parallel steps [36,39], stairs [28,32,36], boulders and rubble [22,56].

Given these advances, the snake's locomotion over large obstacles is still superior, without any visible wobble or slip on high friction steps [12]. This is probably attributed to the animal's more continuous body, additional body compliance in other directions (e.g. rolling, lateral) and ability to actively adjust its body [65] using sensory feedback [66] to conform to the terrain beyond that achievable by passive body compliance. Future studies should elucidate how snakes, and how snake robot should, use tactile sensory feedback control [56–58,60] and combine control compliance [35,58] with mechanical compliance [54,60,67] along multiple directions [52] to stably traverse large, smooth obstacles.

Finally, as our study begins to demonstrate, locomotion in three-dimensional terrain with many large obstacles often involves stochastic transitions, which are statistically affected by locomotor–terrain physical interaction [68–71] (e.g. step height and body compliance here). A new statistical physics view of locomotor transitions [72] will help accelerate the understanding of how animals use or mitigate such statistical dependence and its application in robotic obstacle traversal using physical interaction in the stochastic world.

Data accessibility. This paper has electronic supplementary material, including: materials and methods. Figure S1, experimental set-up and three-dimensional kinematics reconstruction. Electronic supplementary material, figure S2, controller design. Electronic supplementary material, figure S3, effect of body compliance on electrical power. Electronic supplementary material, table S1, sample size. Electronic supplementary material, movie S1, mechanical design of snake robot. Electronic supplementary material, movie S2, snake robot uses a snake-like partitioned gait to traverse a large step rapidly. Electronic supplementary material, movie S3, comparison of large step traversal between rigid and compliant body snake robot. Electronic supplementary material, movie S4, adverse events of snake robot traversing a large step. Excel form S1. Data reported in the paper.

Authors' contributions. Q.F. designed study, developed robot, performed experiments, analysed data, and wrote the paper; C.L. designed and oversaw study and revised the paper. Both authors gave final approval for publication.

Competing interests. The authors declare no competing interests.

Funding. This work was supported by a Burroughs Wellcome Fund Career Award at the Scientific Interface, an Arnold & Mabel Beckman Foundation Beckman Young Investigator award and The Johns Hopkins University Whiting School of Engineering start-up funds to C.L.

Acknowledgements. We thank Hongtao Wu and Zhiyi Ren for building the frame for multi-camera set-up; Tommy Mitchel and Sean Gart for advice on animal data analysis; Nansong Yi and Huidong Gao for help with robot design; Zhiyi Ren for providing initial codes of robot control and Ratan Othayoth, Qihan Xuan, Yuanfeng Han, Yulong Wang and Henry Astley for discussion.

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
