## [Reviewer comments · Royal Society Open Science]

Review History

RSOS-191192.R0 (Original submission)

Review form: Reviewer 1 (Hodjat Pendar)

Is the manuscript scientifically sound in its present form?

No

Are the interpretations and conclusions justified by the results?

Yes

Is the language acceptable?

Yes

Do you have any ethical concerns with this paper?

No

Have you any concerns about statistical analyses in this paper?

Yes

Recommendation?

Accept with minor revision (please list in comments)

Comments to the Author(s)

Please see the attached file (Appendix A).

Review form: Reviewer 2

Is the manuscript scientifically sound in its present form?

Yes

Are the interpretations and conclusions justified by the results?

No

Is the language acceptable?

No

Do you have any ethical concerns with this paper?

No

Have you any concerns about statistical analyses in this paper?

No

Recommendation?

Major revision is needed (please make suggestions in comments)

Comments to the Author(s)

The authors present a snake robot that can climb a step. The videos accompanying the paper are impressive, and do an effective job communicating the main message: a robot with vertical compliance more effectively climbs than a rigid robot. I was particularly impressed with the ability of the robot to recover from errors due to rolling.

The intro and abstract could be improved by focusing on the particular problem at hand. Currently, the problem being solved here is one of traversing a step. However, the intro and abstract barely discuss this. One must wait until section 2 to see the results of the latest state of the art, the author's JEB paper on snakes traversing a step. Moreover, one waits to page 19 to find out that a number of other snake robots have been designed that can traverse a step. Both these sections should be moved to the introduction, because they seem like necessary background for this paper. As a result, the authors will have to delete much of the broad impact statements regarding snake locomotion in the intro. Throughout the paper has a number of issues with presentation.

p.6 line 131 is a run-on sentence and should be re-written. What is a constant, near straight shape?

p.8 The different failure mechanisms are an important part of the results here. I like the attempts to classify them, but its a little unclear what the different failure mechanisms are.

Fig 3 is a good attempt at showing the traversal, but I thought the video did a much clearer job of showing the struggle to climb up the step. The snake does not have very high contrast to the background in the image.

Fig 4 is very confusing. I suggest a before and after picture. The two colors and the dotted lines are not well defined. I understand part part E but parts A-D are confusing.

line 175 needs to be unpacked and rewritten. Many things seem to be listed but I am not clear on what they are and why they could not be overcome.

184 I am still not clear on the difference between a yaw and a roll for this robot. What is so bad about a yaw?

Fig 5 seems very busy and difficult to follow. This figure should be explained. I suggest showing just a single H percent and d going through all the steps. Also, some of the legs do not add up to 1 even when they are coming from the same source. For example, 0.1, 0.5, 0.8 and 0.3 coming out of the imperfect pitch timing.

lines 218-232 should really be in the introduction. I think the authors should say from the beginning that they are going to test compliance.

Fig 6 -- I would have liked this to be combined with Fig 3-- in order to show the improvement of the traversal. This is done nicely in Fig 8

Again Fig 6A is difficult to see. I suggest deleting Fig 3A and 6A and using a schematic to illustrate or some other method.

Fig 7 parts seem quite repetitive in format compared to fig 5. I suggest showing a single step traversal, or moving the entire figure to the Supplement. This figure is not really discussed in the text.

Fig 9a is confusing and needs to be re-drawn. One of the dotted lines is parallel to the leg and the other is not.

How do the authors track whether the wheels are in contact?

Fig 10. Why does the power decrease with step height? that seems counter-intuitive because more gravity is expended to climb higher steps.

Fig 11. I like this figure very much. The authors state that their robot is the fastest, but it seems like T^2 snake is the fastest? The straight lines in Fig11a are not clearly defined. Also, why is the authors' snake the fastest? Is it simply delivering more power? Have a higher power per mass ratio? Or is it using a higher frequency? Stating the rationale for the comparison would help.

366 the authors state that the other robots carefully plan and control their motion to maintain stability. Is there no feedback mechanism in the author's robot? Why is their traversal so much faster

In the discussion, it may be worthwhile stating that the authors have only included one kind of compliance, vertical compliance. Rolling compliance might be even better for preventing rolls. Also, the robots's springs are not clearly analogous to muscles in the snake. Does a snake have energy storage capabilities as well? The snake is covered in muscles in the roll, yaw, and pitch directions, but its not obvious that those muscles can store energy like the robot's springs.

386 -- is there a reason that motors are only actuated at 50 percent? Why not push the limit?

402, it seems strange to compare the speed to a kingsnake given that the robot is bigger and has a different frequency and power output. What is the motivation for comparison to that snake in particular?

408 energy landscape model comes out of nowhere. What is that?

Overall, the writing in this manuscript should be made more concise. The authors' 78 references often make the writing even more difficult to read.

Review form: Reviewer 3

Is the manuscript scientifically sound in its present form?

Yes

Are the interpretations and conclusions justified by the results?

Yes

Is the language acceptable?

Yes

Do you have any ethical concerns with this paper?

No

Have you any concerns about statistical analyses in this paper?

No

Recommendation?

Accept with minor revision (please list in comments)

Comments to the Author(s)

The paper presents some interesting insights about the role of compliance in traversing large steps, via robotic and real snakes. I support its acceptance with just a few minor comments:

"To enable body deformation both laterally and dorsoventrally to achieve similar necessary for traversing large steps"--looks like there's an error in the wording near "similar".

"locked with rotating backward"--when

Fig. 11. I'm surprised by the small values here, just a few percent of body lengths per second. If the body needs to move forward 50% of its length to traverse the step, the king snake requires 25 seconds and the others around one minute. Is this realistic?

Can a kingsnake (or robot) traverse larger steps than shown by using a different gait, that does not involve lateral undulation? How about bending the body into a different shape at the bottom and/or at the top?

Decision letter (RSOS-191192.R0)

03-Sep-2019

Dear Dr Li,

The editors assigned to your paper ("Robotic modeling of snake traversing large obstacles reveals stability benefits of body compliance") have now received comments from reviewers. We would like you to revise your paper in accordance with the referee and Associate Editor suggestions which can be found below (not including confidential reports to the Editor). Please note this decision does not guarantee eventual acceptance.

Please submit a copy of your revised paper before 26-Sep-2019. Please note that the revision deadline will expire at 00.00am on this date. If we do not hear from you within this time then it will be assumed that the paper has been withdrawn. In exceptional circumstances, extensions may be possible if agreed with the Editorial Office in advance. We do not allow multiple rounds of revision so we urge you to make every effort to fully address all of the comments at this stage. If deemed necessary by the Editors, your manuscript will be sent back to one or more of the original reviewers for assessment. If the original reviewers are not available, we may invite new reviewers.

- Data accessibility

<http://datadryad.org/submit?journalID=RSOS&manu=RSOS-191192>

- Competing interests

- Authors' contributions

- Acknowledgements

- Funding statement

Kind regards,

Andrew Dunn

on behalf of Dr Jake Socha (Associate Editor) and Kevin Padian (Subject Editor)

Associate Editor's comments (Dr Jake Socha):

Associate Editor: 1

Comments to the Author:

The reviewers were generally in agreement that this study is worthwhile in advancing our understanding of the mechanics of snake locomotion, using robotic models and comparisons with real snakes. However, there are some major issues that need to be addressed to be considered for publication, which have been identified clearly by the reviewers. We look forward to receiving your revised manuscript.

Comments to Author:

Reviewers' Comments to Author:

Reviewer: 1

Comments to the Author(s)

Please see the attached file.

Reviewer: 2

Comments to the Author(s)

The authors present a snake robot that can climb a step. The videos accompanying the paper are impressive, and do an effective job communicating the main message: a robot with vertical compliance more effectively climbs than a rigid robot. I was particularly impressed with the ability of the robot to recover from errors due to rolling.

The intro and abstract could be improved by focusing on the particular problem at hand. Currently, the problem being solved here is one of traversing a step. However, the intro and abstract barely discuss this. One must wait until section 2 to see the results of the latest state of the art, the author's JEB paper on snakes traversing a step. Moreover, one waits to page 19 to find out that a number of other snake robots have been designed that can traverse a step. Both these sections should be moved to the introduction, because they seem like necessary background for this paper. As a result, the authors will have to delete much of the broad impact statements regarding snake locomotion in the intro. Throughout the paper has a number of issues with presentation.

p.6 line 131 is a run-on sentence and should be re-written. What is a constant, near straight shape?

p.8 The different failure mechanisms are an important part of the results here. I like the attempts to classify them, but its a little unclear what the different failure mechanisms are.

Fig 3 is a good attempt at showing the traversal, but I thought the video did a much clearer job of showing the struggle to climb up the step. The snake does not have very high contrast to the background in the image.

Fig 4 is very confusing. I suggest a before and after picture. The two colors and the dotted lines are not well defined. I understand part part E but parts A-D are confusing.

line 175 needs to be unpacked and rewritten. Many things seem to be listed but I am not clear on what they are and why they could not be overcome.

184 I am still not clear on the difference between a yaw and a roll for this robot. What is so bad about a yaw?

Fig 5 seems very busy and difficult to follow. This figure should be explained. I suggest showing just a single H percent and going through all the steps. Also, some of the legs do not add up to 1 even when they are coming from the same source. For example, 0.1, 0.5, 0.8 and 0.3 coming out of the imperfect pitch timing.

lines 218-232 should really be in the introduction. I think the authors should say from the beginning that they are going to test compliance.

Fig 6 -- I would have liked this to be combined with Fig 3-- in order to show the improvement of the traversal. This is done nicely in Fig 8
Again Fig 6A is difficult to see. I suggest deleting Fig 3A and 6A and using a schematic to illustrate or some other method.

Fig 7 parts seem quite repetitive in format compared to fig 5. I suggest showing a single step traversal, or moving the entire figure to the Supplement. This figure is not really discussed in the text.

Fig 9a is confusing and needs to be re-drawn. One of the dotted lines is parallel to the leg and the other is not.

How do the authors track whether the wheels are in contact?

Fig 10. Why does the power decrease with step height? that seems counter-intuitive because more gravity is expended to climb higher steps.

Fig 11. I like this figure very much. The authors state that their robot is the fastest, but it seems

like T^2 snake is the fastest? The straight lines in Fig11a are not clearly defined. Also, why is the authors' snake the fastest? Is it simply delivering more power? Have a higher power per mass ratio? Or is it using a higher frequency? Stating the rationale for the comparison would help.

366 the authors state that the other robots carefully plan and control their motion to maintain stability. Is there no feedback mechanism in the author's robot? Why is their traversal so much faster

In the discussion, it may be worthwhile stating that the authors have only included one kind of compliance, vertical compliance. Rolling compliance might be even better for preventing rolls. Also, the robots's springs are not clearly analogous to muscles in the snake. Does a snake have energy storage capabilities as well? The snake is covered in muscles in the roll, yaw, and pitch directions, but its not obvious that those muscles can store energy like the robot's springs.

386 -- is there a reason that motors are only actuated at 50 percent? Why not push the limit?

402, it seems strange to compare the speed to a kingsnake given that the robot is bigger and has a different frequency and power output. What is the motivation for comparison to that snake in particular?

408 energy landscape model comes out of nowhere. What is that?

Overall, the writing in this manuscript should be made more concise. The authors' 78 references often make the writing even more difficult to read.

Reviewer: 3

Comments to the Author(s)

The paper presents some interesting insights about the role of compliance in traversing large steps, via robotic and real snakes. I support its acceptance with just a few minor comments:

"To enable body deformation both laterally and dorsoventrally to achieve similar necessary for traversing large steps"--looks like there's an error in the wording near "similar".

"locked with rotating backward"--when

Fig. 11. I'm surprised by the small values here, just a few percent of body lengths per second. If the body needs to move forward 50% of its length to traverse the step, the king snake requires 25 seconds and the others around one minute. Is this realistic?

Can a kingsnake (or robot) traverse larger steps than shown by using a different gait, that does not involve lateral undulation? How about bending the body into a different shape at the bottom and/or at the top?

Author's Response to Decision Letter for (RSOS-191192.R0)

See Appendix B.

RSOS-191192.R1 (Revision)

Review form: Reviewer 1 (Hodjat Pendar)

Is the manuscript scientifically sound in its present form?

Yes

Are the interpretations and conclusions justified by the results?

Yes

Is the language acceptable?

Yes

Do you have any ethical concerns with this paper?

No

Have you any concerns about statistical analyses in this paper?

No

Recommendation?

Accept as is

Comments to the Author(s)

All the issues are addressed properly in the new manuscript.

Review form: Reviewer 2

Is the manuscript scientifically sound in its present form?

Yes

Are the interpretations and conclusions justified by the results?

Yes

Is the language acceptable?

Yes

Do you have any ethical concerns with this paper?

No

Have you any concerns about statistical analyses in this paper?

No

Recommendation?

Accept with minor revision (please list in comments)

Comments to the Author(s)

The Fig 3 is now improved and quite readable. Nice job.

Fig 4 is still confusing, especially parts C and D. Its not clear to the reader what direction yaw and roll are. I suggest using snapshots from the videos like you did in Fig 3. The images C and D just don't work.

Fig 11A -- the gray points are too light to see.

I had already commented that the paper was too long, especially the introduction, but the authors said that the length was justified. There are 11 figures in this paper, and I think at least 3-4 can be moved to supplement. Fig 3 does not use space very well (see the large gap around the middle figure). Fig 5 and 6 seem repetitive. Fig 7 and 10 are only single graphs. This paper is longer than it should be.

Otherwise, I am satisfied with these revisions. Readers will enjoy seeing the snake robots struggling to climb the step, and the use of compliance is a nice way to get the robots to succeed.

Decision letter (RSOS-191192.R1)

17-Dec-2019

Dear Dr Li:

Manuscript ID RSOS-191192.R1 entitled "Robotic modeling of snake traversing large obstacles reveals stability benefits of body compliance" which you submitted to Royal Society Open Science, has been reviewed. The comments of the reviewer(s) are included at the bottom of this letter.

Please submit a copy of your revised paper before 09-Jan-2020. Please note that the revision deadline will expire at 00.00am on this date. If we do not hear from you within this time then it will be assumed that the paper has been withdrawn. In exceptional circumstances, extensions may be possible if agreed with the Editorial Office in advance.

Please note we do not generally allow multiple rounds of revision, so we urge you to make every effort to fully address all of the comments at this stage. If deemed necessary by the Editors, your manuscript will be sent back to one or more of the original reviewers for assessment. If the original reviewers are not available we may invite new reviewers.

- Ethics statement

- Data accessibility

- Competing interests

- Authors' contributions

- Acknowledgements

- Funding statement

Kind regards,

Andrew Dunn

on behalf of Dr Jake Socha (Associate Editor) and Kevin Padian (Subject Editor)

Associate Editor Comments to Author (Dr Jake Socha):

The reviewers agree that the manuscript has been greatly improved, and is now closer to being complete. However, reviewer 2 has some comments that I would like you to address; I agree that

the manuscript is longer than need be, and some (if not all) of the suggested material can be moved to the supplement. If you strongly disagree, please explain why.

Reviewer comments to Author:

Reviewer: 1

Comments to the Author(s)

All the issues are addressed properly in the new manuscript.

Reviewer: 2

Comments to the Author(s)

The Fig 3 is now improved and quite readable. Nice job.

Fig 4 is still confusing, especially parts C and D. Its not clear to the reader what direction yaw and roll are. I suggest using snapshots from the videos like you did in Fig 3. The images C and D just don't work.

Fig 11A -- the gray points are too light to see.

I had already commented that the paper was too long, especially the introduction, but the authors said that the length was justified. There are 11 figures in this paper, and I think at least 3-4 can be moved to supplement. Fig 3 does not use space very well (see the large gap around the middle figure). Fig 5 and 6 seem repetitive. Fig 7 and 10 are only single graphs. This paper is longer than it should be.

Otherwise, I am satisfied with these revisions. Readers will enjoy seeing the snake robots struggling to climb the step, and the use of compliance is a nice way to get the robots to succeed.

Author's Response to Decision Letter for (RSOS-191192.R1)

See Appendix C.

Decision letter (RSOS-191192.R2)

14-Jan-2020

Dear Dr Li:

On behalf of the Editors, I am pleased to inform you that your Manuscript RSOS-191192.R2 entitled "Robotic modeling of snake traversing large, smooth obstacles reveals stability benefits of body compliance" has been accepted for publication in Royal Society Open Science subject to minor revision in accordance with the referee suggestions. Please find the referees' comments at the end of this email.

The reviewers and Subject Editor have recommended publication, but also suggest some minor revisions to your manuscript. Therefore, I invite you to respond to the comments and revise your manuscript.

- Ethics statement

- Data accessibility

<http://datadryad.org/submit?journalID=RSOS&manu=RSOS-191192.R2>

- Competing interests

- Authors' contributions

- Acknowledgements

- Funding statement

Because the schedule for publication is very tight, it is a condition of publication that you submit

the revised version of your manuscript before 23-Jan-2020. Please note that the revision deadline will expire at 00.00am on this date. If you do not think you will be able to meet this date please let me know immediately.

on behalf of Dr Jake Socha (Associate Editor) and Kevin Padian (Subject Editor)
openscience@royalsociety.org

Associate Editor Comments to Author (Dr Jake Socha):

I agree with the authors' reasoning for keeping certain text and figures in the manuscript, and have no other major concerns. Congratulations on a nice piece of work! The only thing that I

noticed (just now, sorry) is that the supplemental movies don't play easily on my Mac. I can open them in VLC, but not either version of Quicktime. If possible, because some in the community are like me, it'd be nice to have the movies in a format that can be easily viewed on PCs or Macs. Perhaps a different codec would help.

Author's Response to Decision Letter for (RSOS-191192.R2)

See Appendix D.

Decision letter (RSOS-191192.R3)

27-Jan-2020

Dear Dr Li,

It is a pleasure to accept your manuscript entitled "Robotic modeling of snake traversing large, smooth obstacles reveals stability benefits of body compliance" in its current form for publication in Royal Society Open Science. The comments of the reviewer(s) who reviewed your manuscript are included at the foot of this letter.

on behalf of Dr Jake Socha (Associate Editor) and Kevin Padian (Subject Editor)
openscience@royalsociety.org

Associate Editor Comments to Author (Dr Jake Socha):
Congratulations again on this article.

Appendix A

Review of ‘Robotic modeling of snake traversing large obstacles reveals stability benefits of body compliance’ by Q. Fu and C. Li

This work is a continuation of a previous work by C. Li and his group on step traverse of kingsnakes (*Lampropeltis mexicana*). Using a robophysical model, it is shown that the compliance can enhance the stability and success rate in traversing steps by increasing/maintaining the contact points with ground. This is a nice study, the experiments are well done, and I believe it is publishable. However, there are a few concerns, mostly minor, that need to be addressed:

- 1- Why the hypotheses are called ‘biological hypotheses’? The experiments are not on a biological system, and the results cannot be directly extended to snakes. I don’t see a significant similarity between the robot and real snake, except they are both elongated bodies. The control system, weight/density per length, power, and even the kinematics of movement are different. Therefore, calling the hypotheses, ‘biological hypotheses’ is a bit stretch. In L78-80, it is not clear that the hypotheses is about snake or robot. For instance hypothesis (2) is: ‘body compliance helps maintain contact and reduce roll instability’. I assume the hypothesis is about the robot. However, because it is called ‘biological hypothesis’ readers may assume this is about animals. This needs to be clarified.
- 2- Serpenoid kinematics (Supplementary Materials) has been used for the parts of the body that are on the horizontal surface (before and after the step).
 - a. For all the trials only one set of parameters (amplitude, phase difference, and wavenumber) is used. I didn’t find any explanation on why serpenoid kinematics is chosen and why these specific parameters? I have not read all of the [14] paper, which is about step traverse of kingsnake. However, only watching the videos of that study, it doesn’t look snakes use merely serpenoid kinematics. Their behavior is more complicated.
 - b. Are the results change (particularly the roll stability) if the kinematic parameters change?
- 3- ‘Good contact’ is used twice in the manuscript. ‘Good’ is a vague word. I’d suggest to change it with a more descriptive term.
- 4- L300-301: ‘below and above the step (gray)’, in the figure the first wheel above the step is in contact, but it is red. Probably ‘gray’ should be changed to ‘solid’.
- 5- I couldn’t understand how the ‘contact probability’ is calculated. L300-302: ‘Contact probability is the ratio of the number wheels in contact with the horizontal surfaces in the body sections below and above the step (gray) to the total number of wheels in these two sections ..’. When the robot moves, the number contact points changes. Is the ‘contact possibility’ measured continuously? Or only a few times during each trial? How the data from different trials are combined?
- 6- L248-249: ‘We speculate that this was due to an increase in energy dissipation from friction within the suspension ..’ What is ‘friction in the suspension’? Energy dissipation could also be due to more stucks.
- 7- L357: ‘.. to understand fundamental principles’, principals of what?
- 8- L361: ‘.. lateral body oscillation necessary to resist large roll instability’, why the lateral body oscillation is ‘necessary’ to resist roll instability? There could be many other ways to overcome the instability and it doesn’t have to be ‘lateral body oscillation’.
- 9- L151: ‘+/-95% confidence interval’ should be ‘95% confidence interval’.

Appendix B

We thank all the reviewers for their comments and revised the paper accordingly. We believe that the paper is greatly improved thanks to the critical feedback.

Major improvements:

1. We moved our recent snake research and previous snake robots traversing steps from other groups to Introduction to provide readers the necessary background. The Introduction section was combined with Section 2 and revised to be more concise.
2. We improved the presentation of the adverse events and their relation to each other. We revised Section 2.4, 2.5, and 3.3 and added more references to corresponding elements of Figs. 5, 6 when introducing adverse events and elaborating more on and how they relate to each other. We revised Fig. 4 and added coloring to Figs. 5, 6 to make them easier to follow. We also added a new Fig. 7 to summarize that roll instability (indicated by roll failure probability) increased with step height but decreased with body compliance.
3. We improved the discussion of how our work advanced snake robot performance traversing large steps (Section 4). We better elaborated why our robot was faster and explained why we chose control parameters and a motor speed of 50% full speed.
4. We revised the entire paper to be more concise, cutting down from 8147 to 7732 words and from 82 to 67 references.

See responses to each comment below.

Associate Editor: 1

The reviewers were generally in agreement that this study is worthwhile in advancing our understanding of the mechanics of snake locomotion, using robotic models and comparisons with real snakes. However, there are some major issues that need to be addressed to be considered for publication, which have been identified clearly by the reviewers. We look forward to receiving your revised manuscript.

RESPONSE:

We thank the reviewers and editor for appreciating our work.

Reviewer: 1

This work is a continuation of a previous work by C. Li and his group on step traverse of kingsnakes (*Lampropeltis mexicana*). Using a robo-physical model, it is shown that the compliance can enhance the stability and success rate in traversing steps by increasing/maintaining the contact points with ground.

This is a nice study, the experiments are well done, and I believe it is publishable.

However, there are a few concerns, mostly minor, that need to be addressed:

1- Why the hypotheses are called ‘biological hypotheses’? The experiments are not on a biological system, and the results cannot be directly extended to snakes. I don’t see a significant similarity between the robot and real snake, except they are both elongated bodies. The control system, weight/density per length, power, and even the kinematics of movement are different. Therefore, calling the hypotheses, ‘biological hypotheses’ is a bit stretch. In L78-80, it is not clear that the hypotheses is about snake or robot. For instance hypothesis (2) is: ‘body compliance helps maintain contact and reduce roll instability’. I assume the hypothesis is about the robot. However, because it is called ‘biological hypothesis’ readers may assume this is about animals. This needs to be clarified.

RESPONSE:

We now simply refer to them as “hypothesis/hypotheses” throughout the paper.

2- Serpenoid kinematics (Supplementary Materials) has been used for the parts of the body that are on the horizontal surface (before and after the step).

a. For all the trials only one set of parameters (amplitude, phase difference, and wavenumber) is used. I didn’t find any explanation on why serpenoid kinematics is chosen and why these specific parameters? I have not read all of the [14] paper, which is about step traverse of kingsnake. However, only watching the videos of that study, it doesn’t look snakes use merely serpenoid kinematics. Their behavior is more complicated.

b. Are the results change (particularly the roll stability) if the kinematic parameters change?

RESPONSE:

We chose serpenoid gait for two reasons:

1. It is close to the animal’s typical lateral oscillation observation from previous animal experiments in [12]. Although kingsnakes showed highly variable waveforms in their body sections below and above the step, to the first order, these two body sections used lateral oscillation (Fig. 3A of [12]).
2. Among the well-studied planar gaits used by snakes, serpenoid gait is one that both well approximates lateral oscillation and is easy to implement in snake robots, e.g.:
 - Hirose S. 1993 *Biologically inspired robots: snake-like locomotors and manipulators*. Oxford University Press.
 - Transth A A, Liljebäck P and Pettersen K Y 2007 Snake robot obstacle aided locomotion: An experimental validation of a non-smooth modeling approach *IEEE Int. Conf. Intell. Robot. Syst.* 2582–9.
 - Tanaka M and Tanaka K 2015 Control of a Snake Robot for Ascending and Descending Steps *IEEE Trans. Robot.* 31 511–20.

Because our study did not focus on gait optimization but rather on stability principles, serpenoid gait was chosen for simplicity.

In this study, we chose serpenoid wave parameters from preliminary experiments and kept them fixed to test the effect of body compliance and step height. Wave parameter variation will likely quantitatively affect roll stability. For example, higher wave amplitude of lateral oscillation will increase roll stability. However, we do not expect that the dependence on step height and body compliance to change for any given wave parameter, i.e., roll stability will decrease with step height and increase with body compliance.

Revised.

Lines 121-124:

“Serpentoid traveling wave was selected because it is a planar laterally oscillating pattern similar to that used by kingsnakes and easy to implement in snake robots [18,45,46]. We chose wave parameters from preliminary experiments and kept them fixed in this study to test the effect of step height and body compliance.”

Lines 370-373:

“We note that our robot still has potential to achieve even higher speeds at high traversal probability, because in our experiments the motors were actuated at only 50% full speed to protect the robot and we have yet to systematically test and identify optimal serpentoid wave parameters [57].”

3- ‘Good contact’ is used twice in the manuscript. ‘Good’ is a vague word. I’d suggest to change it with a more descriptive term.

RESPONSE:

Removed “good”.

4- L300-301: ‘below and above the step (gray)’, in the figure the first wheel above the step is in contact, but it is red. Probably ‘gray’ should be changed to ‘solid’.

RESPONSE:

Revised.

Lines 300-303:

“Contact probability is the ratio of number wheels in contact with the horizontal surfaces in body sections below and above step to total number of wheels in these two sections. Body sections below and above the step are shown in gray, and cantilevering body section is shown in red. Solid wheels are in contact with the surface and dashed wheels are not.”

5- I couldn’t understand how the ‘contact probability’ is calculated. L300-302: ‘Contact probability is the ratio of the number wheels in contact with the horizontal surfaces in the body sections below and above the step (gray) to the total number of wheels in these two sections ..’. When the robot moves, the number contact points changes. Is the ‘contact possibility’ measured continuously? Or only a few times during each trial? How the data from different trials are combined?

RESPONSE:

We revised the main text and Fig. 8 caption to briefly summarize this and provide an example. We also referred the readers to Supplementary Material, where this has been explained in detail (Lines 186-190).

Lines 296-297:

“Both were averaged spatiotemporally over the traversal process across all pitch segments in these two sections together for each trial.”

Line 303:

“In this example, contact probability = $3/7 = 43\%$.”

Supplementary material lines 186-190:

“Contact probability, deformation, and surface conformation difference were averaged spatiotemporally over time and across all pitch segments in the section above and below the step for each trial. Electrical power was averaged over time for each trial. Finally, these trial averages were further averaged across trials for each step height and body compliance treatment to obtain treatment means and standard deviations (s.d.), which are reported in all figures.”

6- L248-249: ‘We speculate that this was due to an increase in energy dissipation from friction within the suspension ..’ What is ‘friction in the suspension’? Energy dissipation could also be due to more stucks.

RESPONSE:

Revised.

Lines 349-352:

“We speculate that this was due to an increase in energy dissipation from the motors holding their positions to maintain body configuration, friction against the surfaces due to higher contact probability, viscoelastic response [51] of the suspension, and motor stalling and wheel sliding when the robot was stuck.”

7- L357: ‘.. to understand fundamental principles’, principals of what?

RESPONSE:

Revised.

Line 362:

“Our study also advanced the performance of snake robots traversing large steps.”

8- L361: ‘.. lateral body oscillation necessary to resist large roll instability’, why the lateral body oscillation is ‘necessary’ to resist roll instability? There could be many other ways to overcome the instability and it doesn’t have to be ‘lateral body oscillation’.

RESPONSE:

Revised.

Lines 44-45:

“Surmounting large, smooth obstacles like steps has often been achieved by using a simple follow-the-leader gait [31–41], in which the body mainly deforms within a vertical plane with little lateral deformation and a narrow base of ground support.”

9- L151: ‘+/-95% confidence interval’ should be ‘95% confidence interval’.

RESPONSE:

Fixed.

Reviewer: 2

The authors present a snake robot that can climb a step. The videos accompanying the paper are impressive, and do an effective job communicating the main message: a robot with vertical compliance more effectively climbs than a rigid robot. I was particularly impressed with the ability of the robot to recover from errors due to rolling.

The intro and abstract could be improved by focusing on the particular problem at hand. Currently, the problem being solved here is one of traversing a step. However, the intro and abstract barely discuss this. One must wait until section 2 to see the results of the latest state of the art, the author's JEB paper on snakes traversing a step. Moreover, one waits to page 19 to find out that a number of other snake robots have been designed that can traverse a step. Both these sections should be moved to the introduction, because they seem like necessary background for this paper. As a result, the authors will have to delete much of the broad impact statements regarding snake locomotion in the intro.

RESPONSE:

As suggested, we moved the parts reviewing our recent animal study and previous snake robots traversing steps to Introduction to provide readers the necessary background as suggested. We also revised Introduction so it is more concise.

However, we kept the first two paragraphs to provide context how this research can potentially have a broader impact. The example of large step traversal that we focused on is highly relevant to the broader scenarios. In addition, we have been very clear in drawing conclusions based on the data and not over-stating in the last section. Further, our Introduction is only ~two pages out of a ~20 page paper (main text). Thus, we believe that the broader introduction is well justified.

Throughout the paper has a number of issues with presentation.

p.6 line 131 is a run-on sentence and should be re-written. What is a constant, near straight shape?

RESPONSE:

Revised.

Lines 125-130:

“To generate cantilevering on the section in between, we used the minimal number of segments required to bridge across the step. The cantilevering section was straight and as vertical as possible, other than the two pitch segments near the section above the step bending down for proper contact with the upper surface (see details in Supplementary Material). This shape was calculated based on the step height measured from online camera tracking before body cantilevering started and remained the same while traveling down the body.”

Fig 3 is a good attempt at showing the traversal, but I thought the video did a much clearer job of showing the struggle to climb up the step. The snake does not have very high contrast to the background in the image.

RESPONSE:

We increased the contrast of the snake robot relative to the background (using GIMP software) to better highlight the robot in Fig. 3.

p.8 The different failure mechanisms are an important part of the results here. I like the attempts to classify them, but its a little unclear what the different failure mechanisms are.

We revised Fig. 4 and Section 2.4 to better illustrate each failure mechanism and how they relate to each other. See detailed responses to the comments below.

Fig 4 is very confusing. I suggest a before and after picture. The two colors and the dotted lines are not well defined. I understand part E but parts A-D are confusing.

RESPONSE:

We separated Fig. 4A-D into before and after schematics. This allowed us to remove unnecessary dots and dashed lines.

line 175 needs to be unpacked and rewritten. Many things seem to be listed but I am not clear on what they are and why they could not be overcome.

RESPONSE:

Revised.

Lines 164-172:

“(1) Imperfect pitch timing (Fig. 4A-B; Fig. 5, gray). This includes pitching early (Fig. 4A) or pitching late (Fig. 4B) due to inaccurate estimation of body forward positions relative to the step. Noise throughout the system, both mechanical (e.g., variation in robot segment construction and surface friction) and in feedback control (e.g., camera noise, controller delay), resulted in large trial-to-trial variation of robot motion and interaction with the step, leading to inaccurate estimation. With underestimation, segments still far away from the step was raised (pitched down) too early (Fig. 4A). With overestimation, segments close the step was raised (pitched down) too late and pushed against the vertical surface of the step (Fig. 4B). These control imperfections often triggered other involuntary adverse events, as described below.”

184 I am still not clear on the difference between a yaw and a roll for this robot. What is so bad about a yaw?

RESPONSE:

Yawing is rotating mainly around a vertical axis, whereas rolling is rotating mainly around the fore-aft axis of the entire body.

Yawing itself does not cause failure, but it may trigger early pitch timing or rolling.

Revised:

Lines 178-181:

“Because of this, small lift-off and slip of the segments in contact with the horizontal surfaces often accompanied yawing, which could lead to rolling described below (Fig. 5, blue arrows). Yawing could also compromise segment position estimation and lead to further imperfect pitch timing (Fig. 5, arrows from blue to gray box).”

Lines 193-195:

“(3) Rolling (Fig. 4D; Fig. 5, red). The robot rolled around the fore-aft axis (wobbled) substantially when the center of mass (Fig. 4D, black point) projection (gray point) moved out of the base of support (orange shade).”

Fig 5 seems very busy and difficult to follow. This figure should be explained. I suggest showing just a single H percent and d going through all the steps. Also, some of the legs do not add up to 1 even when they are coming from the same source. For example, 0.1, 0.5, 0.8 and 0.3 coming out of the imperfect pitch timing.

RESPONSE:

We revised Sections 2.4, 2.5, and 3.3 and added more references to elements of Figs. 5 and 6 when introducing the adverse events and how they relate to each other. We also added color to several more important transition arrows in Fig. 5 that are discussed in texts to highlight them.

We kept the original structure of Figs. 5 and 7 (now Figs. 5 and 6). We strongly feel that this diagram is useful by not only demonstrating the complexity and stochasticity of transitions between events but also highlighting how the trends statistically depend on step height and body compliance.

We note that our transition diagram is not the same as a Markov chain. Here, to show the frequency of each transition behavior relative to the entire sample, we define probability of occurrence as the ratio of the number of trials with occurrence to the total number of trials. Because an adverse event can occur multiple times in a trial, the probability of occurrence of each arrow may not add up to 1 given our definition. However, for each event except start and end (including succ. and fail.), the sum of probabilities of arrows coming out of it equals to that of arrows going into it. We have double-checked calculations.

Revised Figs. 5 and 6 captions:

Lines 216-219:

“Each arrow is a transition between events, with arrow thickness proportional to its probability of occurrence, shown by number next to it. Note that probability of occurrence here is ratio of number of trials with occurrence to total number of trials and is different from transition probability in Markov chains.”

lines 218-232 should really be in the introduction. I think the authors should say from the beginning that they are going to test compliance.

RESPONSE:

Revised.

Lines 70-79:

“In this study, we take the next step in understanding the stability principles of large step traversal using body partitioning with large lateral oscillation, by testing two hypotheses: (1) roll stability diminishes as step height increases; and (2) body compliance helps maintain contact and reduce roll instability. The snakes did not attempt to traverse larger steps and it was impractical to directly modify their body compliance without affecting locomotion. Thus, in order to test these two hypotheses, we developed a snake robot as a physical model which we could systematically challenge with increasingly high steps and whose body compliance could be modified. The second hypothesis was motivated by the observation during preliminary experiments that the snake robot

with rigid body often rolled to the extent of involuntary lift-off from horizontal surfaces, in contrast to the snake with compliant body [2] that never did so [12] (see Section 2.5 for more detail).”

Fig 6 -- I would have liked this to be combined with Fig 3-- in order to show the improvement of the traversal. This is done nicely in Fig 8

RESPONSE:

Combined.

Again Fig 6A is difficult to see. I suggest deleting Fig 3A and 6A and using a schematic to illustrate or some other method.

RESPONSE:

We increased the contrast of the snake robot relative to the background (using GIMP software) to better highlight the robot in Fig. 3 (now combined for both rigid and compliant body).

Fig 7 parts seem quite repetitive in format compared to fig 5. I suggest showing a single step traversal, or moving the entire figure to the Supplement. This figure is not really discussed in the text.

RESPONSE:

See response to comment about Fig. 5 above.

Fig 9a is confusing and needs to be re-drawn. One of the dotted lines is parallel to the leg and the other is not.

How do the authors track whether the wheels are in contact?

RESPONSE:

The two dotted lines had different directions according to our previous different definition of surface conformation of the wheels in contact and not in contact. To avoid confusion, we redefined them to have the same direction and revised the text accordingly.

Lines 318-320:

“Surface conformation was defined as the positive or negative minimal distance from the surface of a wheel contacting or lifted off the surface (Fig. 9A inset, right or left wheel).”

Tracking of wheel contact was included in Supplementary Material, and we revised Lines 293-297 to briefly explain this and guide readers interested in the calculation to a detailed explanation in supplementary material.

Lines 293-297:

“Both wheel contact and body deformation were determined by examining whether any part of the wheel penetrated the step surface assuming no suspension compression, based on 3-D reconstruction of the robot from high speed videos. Both were averaged spatiotemporally over the traversal process across all pitch segments in these two sections together for each trial. See details in Supplementary Material”

Fig 10. Why does the power decrease with step height? that seems counter-intuitive because more gravity is expended to climb higher steps.

RESPONSE:

We speculate that the decrease of power consumption with step height may result from the decrease in the number of laterally oscillating segments, which dissipated energy during sliding against the surfaces.

We added a sentence in Lines 352-354 to briefly explain this.

Lines 352-354:

“The decrease of power consumption with step height may result from the decrease in the number of laterally oscillating segments, which dissipated energy during sliding against the surfaces. In addition, the electrical energy consumed was two orders of magnitude larger than the mechanical work needed to lift the body onto the step; the majority of the energy was not used to do useful work.”

Fig 11. I like this figure very much. The authors state that their robot is the fastest, but it seems like T² snake is the fastest? The straight lines in Fig11a are not clearly defined. Also, why is the authors' snake the fastest? Is it simply delivering more power? Have a higher power per mass ratio? Or is it using a higher frequency? Stating the rationale for the comparison would help.

RESPONSE:

We actually did not state that our robot is the fastest, but rather faster than “most previous snake robots” (Line 364). We also had a sentence stating why T² Snake-3 is faster (Lines 368-370).

We moved the definition of straight lines in Fig. 11A from after (B) to after (A) so it is easier to see.

We recognize the reviewer’s suggestions. In fact, we have wondered about these possibilities. However, very few detailed specifications reported in previous snake robots study, and we could not compare power, power mass ratio, motor speed, frequency, etc. Thus, we Reasons of why our robot is faster are in the response to the next comment.

Lines 378-379:

“Speeds of previous robots are the fastest reported values from [28,31,35–38,40] or accompanying videos.”

366 the authors state that the other robots carefully plan and control their motion to maintain stability. Is there no feedback mechanism in the author's robot? Why is their traversal so much faster

RESPONSE:

There is a simple feedback logic controller in our robot which controlled section division propagation only.

However, our robot only relied on the lateral oscillation in the body sections above and below the step and body compliance to resist instability. This saved lots of computational resources used on

other robots for complex feedback control and thus increased the loop frequency of the entire system. We revised Lines 370-372 accordingly.

Revised:

Lines 366-368:

“These improvements were attributed to the inherent roll stability from large lateral oscillation and an improved ability to maintain terrain contact via body compliance while using a simple, faster controller.”

Lines 48-49:

“Regardless, all these previous snake robots rely on careful planning and control of their motion to maintain stability and thus often traverses at low speeds.”

In the discussion, it may be worthwhile stating that the authors have only included one kind of compliance, vertical compliance. Rolling compliance might be even better for preventing rolls. Also, the robots's springs are not clearly analogous to muscles in the snake. Does a snake have energy storage capabilities as well? The snake is covered in muscles in the roll, yaw, and pitch directions, but its not obvious that those muscles can store energy like the robot's springs.

RESPONSE:

Revised.

Lines 394-397:

“Although our discoveries were made on simple large steps with only vertical body compliance, the use of lateral body oscillation and body compliance to achieve a large, reliable base of support for roll stability may be broadly useful for snakes and snake robots traversing other large, smooth obstacles in terrain like non-parallel steps [36,39], stairs [28,32,36], boulders, and rubble [22,53].”

Our study only showed that the suspension springs allowed better conforming of the snake robot with the step surfaces, which improved roll stability. It is not clear whether the springs functioned as an energy-saving spring in our robot, or whether snake muscles can do so.

Lines 400-402:

“This is likely attributed to the animal’s more continuous body, additional body compliance in other directions (e.g., rolling, lateral), and ability to actively adjust the body [60] to conform to the terrain beyond achievable by passive body compliance.”

386 -- is there a reason that motors are only actuated at 50 percent? Why not push the limit?

RESPONSE:

We only actuated motors at 50% full speed in experiments to protect the robot. Running the motors at the full speed makes it difficult for the experimenter to respond in time when the robot fell off the step, which often resulted in damage and the need for repair. In addition, with 19 segments in the robot and up to 10 to raise during initial cantilevering, running the motors at the full speed may exceed their power capability and trigger current protection which shuts down the motors.

Revised:

Lines 370-373:

“We note that our robot still has potential to achieve even higher speeds at high traversal probability, because in our experiments the motors were actuated at only 50% full speed to protect the robot and we have yet to systematically test and identify optimal serpenoid wave parameters [57].”

402, it seems strange to compare the speed to a kingsnake given that the robot is bigger and has a different frequency and power output. What is the motivation for comparison to that snake in particular?

RESPONSE:

We included the snake to provide a baseline of high performance that the robots aspire to achieve.

We have normalized as much as we can to compare across all the robots and the snake. We only considered vertical traversal speed as some robots and the snake uses lateral oscillation while others did not. We also normalized vertical speed to body length. As mentioned in the response to the comment about Fig. 11 above, because previous studies did not report parameters like frequency and power, we could not do a better comparison.

408 energy landscape model comes out of nowhere. What is that?

RESPONSE:

Deleted.

Overall, the writing in this manuscript should be made more concise. The authors' 78 references often make the writing even more difficult to read.

RESPONSE:

We revised the entire paper to be more concise, cutting down from 8147 to 7732 words and from 82 to 67 references.

Reviewer: 3

The paper presents some interesting insights about the role of compliance in traversing large steps, via robotic and real snakes. I support its acceptance with just a few minor comments:

"To enable body deformation both laterally and dorsoventrally to achieve similar necessary for traversing large steps"--looks like there's an error in the wording near "similar".

RESPONSE:

Fixed.

"locked with rotating backward"—when

RESPONSE:

Fixed.

Fig. 11. I'm surprised by the small values here, just a few percent of body lengths per second. If the body needs to move forward 50% of its length to traverse the step, the king snake requires 25 seconds and the others around one minute. Is this realistic?

RESPONSE:

We double-checked that all the calculations are correct.

The traversal speed in Fig. 11B is vertical speed. During traversal of a large step, a snake robot or snake needs to not only gain height but also move forward.

For example, the kingsnake needs to move forward by ~ 0.7 body length to traverse a step as high as 12.5% body length. At the typical horizontal speed observed in our previous animal study, this alone takes 2 s. Even if gaining vertical height does not slow the animal down in the forward direction, this alone results in a vertical speed of only 6.25% body length/s. In reality, the animal does slow down in the forward direction in order to gain vertical height, which results in an even smaller actual vertical speed of 2% body length/s. This same analysis applies to the robot, resulting in a small vertical speed during large step traversal. In addition, the lifting motion requires the motor to generate large torques and decreases motor speed given the same power. Thus, the small vertical speeds are not surprising.

We added “vertical” to Fig. 11B y-label and revised Fig. 11 caption to make this more clear.

Lines 381-382:

“Note that vertical traversal speed, i.e., normalized step height divided by traversal time, is the slope of lines connecting each data point to the origin in (A).”

Can a kingsnake (or robot) traverse larger steps than shown by using a different gait, that does not involve lateral undulation? How about bending the body into a different shape at the bottom and/or at the top?

RESPONSE:

Our previous animal study observed that kingsnakes always used lateral oscillation in combination with cantilevering to traverse large steps, regardless of step height and surface friction. We did not observe it using other distinct gaits without lateral oscillation.

Some snake robots indeed used other gaits that did not involve lateral oscillation to traverse large steps. We have discussed them in comparison with our robot.

Lines 52-54:

“The body sections below and above the step always oscillate laterally on the horizontal surfaces to propel forward, while the body section in between cantilevers in the air in a vertical plane to bridge the large height increase.”

Lines 44-48:

“Surmounting large, smooth obstacles like steps has often been achieved by using a simple follow-the-leader gait [31–41], in which the body mainly deforms within a vertical plane with little lateral deformation and a narrow base of ground support. Only two previous snake robots deliberately

used lateral body deformation to provide a wide base of support when traversing large steps [28,39,40].”

Appendix C

Dear Editor (Jake),

We thank both reviewers for taking the time to read our paper and providing helpful feedback.

Considering Reviewer 2's comments, we merged Fig. 7 into Fig. 3 to better use space and moved Fig. 10 to supplementary material, reducing the number of figures from 11 to 9.

We strongly feel that moving Figs. 3C, 5, 6, and 7 would subtract from the paper. We elaborated our reasoning below.

However, thanks to Reviewer 2's comments, we now better emphasized the stochastic nature of transitions among adverse events that lead to failure and the statistical nature of the improvements in surface contact by adding body compliance. This helps the readers better appreciate this key result of our work. We attached a PDF with these changes highlighted.

In addition, we proofread and polished the entire paper (including supplementary material).

Below are point-by-point responses.

Associate Editor Comments to Author (Dr Jake Socha):

The reviewers agree that the manuscript has been greatly improved, and is now closer to being complete. However, reviewer 2 has some comments that I would like you to address; I agree that the manuscript is longer than need be, and some (if not all) of the suggested material can be moved to the supplement. If you strongly disagree, please explain why.

Reviewer comments to Author:

Reviewer: 1
Comments to the Author(s)

All the issues are addressed properly in the new manuscript.

RESPONSE:

We thank the reviewer for helping us improve our paper.

Reviewer: 2
Comments to the Author(s)

The Fig 3 is now improved and quite readable. Nice job.

RESPONSE:

We thank the reviewer for the comment.

Fig 4 is still confusing, especially parts C and D. Its not clear to the reader what direction yaw and roll are. I suggest using snapshots from the videos like you did in Fig 3. The images C and D just don't work.

RESPONSE:

Thanks for the suggestion. We changed the figure using snapshots. See the new Fig. 4D-E.

Fig 11A -- the gray points are too light to see.

RESPONSE:

We made it darker. See the new Fig. 9.

I had already commented that the paper was too long, especially the introduction, but the authors said that the length was justified.

RESPONSE:

We maintain our opinion that introduction is not too long.

Lengthwise, it is just over 2 pages (including a figure) as compared to results (Sections 2-4) which are over 16 pages.

More importantly, all the four paragraphs are necessary to introduce the intended scientific contributions. The first two paragraphs of Intro give broader introduction and motivation on the big question we are addressing for snake and snake robots, respectively. The third paragraph provides context of our previous animal study and define the model system, which are required to introduce hypothesis to be tested. The fourth paragraph proposes our hypotheses and approach to testing them.

There are 11 figures in this paper, and I think at least 3-4 can be moved to supplement. Fig 3 does not use space very well (see the large gap around the middle figure). Fig 5 and 6 seem repetitive. Fig 7 and 10 are only single graphs. This paper is longer than it should be.

RESPONSE:

We thank the reviewer for the continued suggestion to improve our paper.

We made the following improvements:

1. To better use space, we combined Fig. 7 into Fig. 3 (now panel D).
2. We moved Fig. 10 to supplementary material.

Together, these reduced the number of figures from 11 to 9.

However, we strongly feel that moving Figs. 3C, 5, 6, and 7 to supplementary material would subtract from the main messages of the paper.

Fig. 3C and Fig. 7 are part of the main results, showing that traversal performance (Fig. 3C) and stability (Fig. 7) diminish as step becomes higher, but they are improved by adding body compliance. These results are key in addressing our hypotheses and clearly belong to the main text.

Fig. 5 and 6 are not repetitive. They are for the rigid and compliant body robot, respectively. These transition pathways, and how they change as body compliance is introduced, are a key result of the paper. They highlight that locomotion in such 3-D terrain with large obstacles (even for a simple representative case of step obstacles that we tested) are highly stochastic. Given the stochastic nature, statistical trends emerge as step height increases. In addition, it is by changing the statistical trends of these transition pathways that the introduction of body compliance improved stability and traversal probability. We feel that, although these statistical quantification seem complex at first glance, they add new insight and can suggest ideas that are simple and potentially universal to the field. Moving them to supplementary material would subtract from the paper.

However, thanks to Reviewer 2's comments, we now better emphasized the stochastic nature of transitions among adverse events that lead to failure and the statistical nature of the improvements in surface contact by adding body compliance. This helps the readers better appreciate this key result of our work. We highlighted these changes on the PDF of the paper uploaded.

Otherwise, I am satisfied with these revisions. Readers will enjoy seeing the snake robots struggling to climb the step, and the use of compliance is a nice way to get the robots to succeed.

RESPONSE:

We thank the reviewer for appreciating the value of our work and helping us improve our paper.

Appendix D

Dear Editor (Jake),

Thank you for spotting this issue.

We have converted the movies to .MOV format with H.264 codec.

We have also followed instructions to add end statements and upload files for typesetting.

Thank you and the reviewers again for appreciating our work and helping us improving our paper.

Best regard,

Chen Li